# Conductive Silver/Carbon Fiber Films for Rapid Detection of Human Coronavirus

**DOI:** 10.3390/polym14101983

**Published:** 2022-05-12

**Authors:** Hwan Gyun Jeon, Ji Wook Choi, Hee Uk Lee, Bong Geun Chung

**Affiliations:** Department of Mechanical Engineering, Sogang University, Seoul 04107, Korea; switchgyun@gmail.com (H.G.J.); delete1821@gmail.com (J.W.C.); fluxion78@gmail.com (H.U.L.)

**Keywords:** silver/carbon fiber, low power thermal cycler, polymerase chain reaction, human coronavirus

## Abstract

Polymerase chain reaction has gained attention since the outbreak of novel coronavirus in 2019. Due to its high specificity and capability for early detection, it is considered a standard method for the diagnosis of infectious diseases. However, the conventional thermocyclers used for nucleic acid amplification are not suitable for point-of-care testing applications, as they require expensive instruments, high-power consumption, and a long turnaround time. To suppress the widespread of the pandemic, there is an urgent need for the development of a rapid, inexpensive, and portable thermal cycler. Therefore, in this paper, we present a conductive silver/carbon fiber film-based thermal cycler with low power consumption (<5 W), efficient heating (~4.5 °C/s), low cost (<USD 200), and handheld size (11.5 × 7.1 × 7.5 mm). The conductive film, which was used as a heating source of the thermal cycler, was fabricated by the electrochemical deposition method. The successful coating of Ag was characterized by a scanning electron microscope and confirmed by energy-dispersive X-ray spectroscopy. The film showed excellent electrical/thermal conductivity and durability. Using our thermal cycler, 35 cycles of amplification were accomplished within 10 min. We also successfully demonstrated the multiplexed detection of various human coronaviruses (e.g., OC43, 229E, and NL63) using our thermal cycler.

## 1. Introduction

Since the outbreak of the coronavirus disease 2019 (COVID-19) pandemic, caused by the severe acute respiratory syndrome coronavirus-2 (SARS-CoV-2), diagnosis technologies have been receiving a great deal of attention [1,2]. As early detection is vital to prevent the rapid spread of the pandemic, the nucleic acid amplification tests (NAATs) tests are considered the gold standard method for clinical diagnosis of COVID-19 [3,4]. In particular, polymerase chain reaction (PCR) is employed as the primary method due to its single-molecule sensitivity and excellent specificity [5]. PCR requires repeated thermal cycling with two or three temperatures to amplify the target molecules. For thermal cycling, the conventional PCR systems generally use a Peltier-based heater and require an hour and more for 30–40 cycles of thermal cycling [6]. Although a number of improvements have been made in the systems, the slow ramping rate of the thermal cycler and long processing time are still critical limitations [7,8,9,10]. These restrictions have become major hurdles for their on-site applications, especially in a resource-limited setting [11].

To address these limitations, fast PCR thermal cyclers, such as convective heating [12,13], photonic PCR [14,15], and resistive heating [16,17], have previously been developed to reduce amplification time. Convective heating is generally used to perform PCR thermal cycling within a short time [18]. Convective PCR thermal cycling, in which PCR is conducted by inducing spontaneous thermal convection inside a capillary tube, is a promising tool due to its simple heating strategy and low cost. However, the non-uniformity of the temperature is still a challenge, because the sample unit is extremely sensitive to temperature fluctuations [19]. Photonic PCR, which is based on the photothermal effects of nanomaterials (e.g., gold nanoparticles, magnetic nanoparticles, and graphene), is another strategy to enhance the heating rate of thermal cycling. Photothermal nanomaterials used as light-to-heat converters enable excellent heating rates and uniform temperature gradients due to their superior thermal conductivity and volumetric heating [20]. Despite these advantages, the high cost and mechanical weakness of nanomaterials limit their actual applications [21]. In addition, the photobleaching caused by the overlap of wavelength bands between the light source and the probe can be a complicated issue for fluorescence detection of PCR products [22]. Resistive heating may be an alternative method to resolve the abovementioned limitations. By integrating a microfabricated thin-film heater, thermal cycling time can be reduced, due to its high heat transfer rate and small sample volume [23]. Resistive heating also shows stable heating performance due to its excellent electrical/thermal stability [24]. For these reasons, conductive materials, such as copper, chromium, and platinum, have been employed as heating elements [25]. However, these materials are not suitable for point-of-care testing (POCT) applications due to complex fabrication processes. There is an urgent need of the fast, cheap, low-power, and portable PCR thermal cyclers for POCT diagnostics.

In this paper, we developed the novel conductive silver/carbon fiber (Ag/CF)-film-based thermal cycler with low cost, portable size, low-power consumption, and efficient heating rate. The Ag/CF film used as a heating source was simply fabricated by dropping silver carbonate solutions onto a commercial CF. The successful coating and characterization of Ag were confirmed with scanning electron microscopy (SEM) and energy-dispersive X-ray spectroscopy (EDS). The optimized Ag/CF film showed excellent electrical/thermal conductivity, and PCR thermal cycling time was reduced to 10 min. Furthermore, the multiplexed detection of the human coronaviruses (e.g., OC43, 229E, and NL63) was demonstrated using our thermal cycler.

## 2. Materials and Methods

### 2.1. Fabrication of Conductive Ag/CF Film

For the fabrication of the Ag/CF film, a silver conductive solution was prepared as previously described [26]. Briefly, 20 mL of methanol (Sigma Aldrich, USA) and 10.8 g of 2-Amino-2-methyl-1-propanol (AMP, Sigma Aldrich, St. Louis, MO, USA) were mixed, and then 3.2 g of silver carbonate powder (Ag_2_CO_3_, Sigma Aldrich, St. Louis, MO, USA) was dispersed in the mixture. The mixture was stirred with a magnetic stirrer for 2 h at 30 °C, sonicated for 10 min, and stored at 4 °C for further use. The commercial CF (FRP shop, Seoul, Korea) was fixed on the thin glass with a double-sided carbon tape. Oxygen plasma treatment (Femto Scientific, Hwaseong, Korea) was performed to enhance the interfacial adhesion of the CF surface. For this procedure, 200 μL of prepared Ag_2_CO_3_ solution was evenly dropped on the CF surface via manual pipetting. The resulting Ag/CF film was dried and reacted with a two-step thermal treatment using a drying oven (DAIHAN, Wonju, Korea).

### 2.2. Characterization of Ag/CF Film

To analyze the surface morphological characteristics of the Ag/CF film, SEM (JSM-7100F, JEOL, Tokyo, Japan) was employed. The samples were prepared on a 5 mm × 5 mm Si wafer with 5 µL of Ag_2_CO_3_ solution. EDS was performed using the same equipment to confirm the elemental composition of the Ag/CF film. The EDS mapping image was analyzed using image processing software (Image J, NIH, Bethesda, MD, USA). The sheet resistance of the Ag/CF film was measured by a system source meter (4200A-SCS, Keithley, Cleveland, OH, USA). Thermal images were taken with an infrared camera (E60, FLIR Systems Inc., Wilsonville, OR, USA) and analyzed using a software program provided by the manufacturer (FLIR tools, FLIR Systems Inc., Wilsonville, OR, USA). Thermogravimetry analysis (TGA) of the film was conducted with a thermal analysis system (TGA Q50, Ta Instruments, New Castle, DE, USA), under the following conditions: 10 mg, nitrogen atmosphere (60 mL/min) temperature range 30 °C to 600 °C, a heating rate of 20 °C/min.

### 2.3. Assembly of PCR Thermal Cycler

The Ag/CF-film-based thermal cycler consisted of a single board computer (Raspberry pi 4B, Adafruit, New York, NY, USA), temperature sensor (MLX90614, Melexis, Belgium), a relay module (DFR0473, DFRobot, Shanghai, China), and cooling fan (LD3007MS, ICBanq, Seoul, Korea). The external housing and internal mount were designed with a three-dimensional (3D) CAD software (Inventor, Autodesk, San Rafael, CA, USA) and printed using a 3D printer (DP200, Sindoh, Seoul, Korea). The control modules were connected to general-purpose input–output (GPIO) pins of Raspberry pi through a circuit board. The metal-oxide-semiconductor field-effect transistor (MOSFET, SK C3851 67 Y, SK Hynix, Icheon, Korea) was mounted on the circuit board to control the cooling fan. The temperature sensor was assembled to measure temperature in real time. The Raspberry pi was powered by a USB-C type power supply (Adafruit, New York, NY, USA).

### 2.4. PCR Experiments

PCR samples were prepared in a 50 µL of master mix containing 1 µL of the template DNA, 1 µL of forward primer (10 µM), 1 µL of reverse primer (10 µM), and 1 µL of probes (10 µM) following the manufacturer’s protocol. The detailed sequences are shown in Appendix A. Polydimethylsiloxane (PDMS, Dow Corning, Midland, MI, USA) chip was fabricated for PCR applications. PDMS was mixed in a ratio of 10:1 (base:curing agent), poured onto the Si wafer, and baked at 85 °C for 2 h. The PDMS was cut into a size of 20 × 20 mm and punched to form chambers. The PDMS mold was bonded onto a thin slide glass after oxygen plasma treatment. Then, 10 µL samples were pipetted into the chambers and covered with 10 µL of mineral oil (Sigma Aldrich, St. Louis, MO, USA). The chip was placed in the center of the Ag/CF film, and thermal cycling was conducted using the control modules. The PCR protocol consisted of 35 cycles of denaturation for 5 s at 95 °C and annealing–extension for 30 s at 58 °C. Subsequently, the imaging of the PDMS chip was carried out using a fluorescence microscope (IX37, Olympus, Shinjuku, Japan). The intensities of the fluorescence images were analyzed by ImageJ software.

## 3. Results and Discussion

### 3.1. Ag/CF-Film-Based Portable PCR Thermal Cycler

We developed a portable Ag/CF-film-based PCR thermal cycler for rapid detection of human coronavirus (Figure 1). Ag/CF films were fabricated by the electrochemical deposition method (Figure 1a). Silver carbonate was used for the precursor, as the silver content was higher than silver citrate and silver nitrate [26]. Before coating, the surface of CF was treated with oxygen plasma to increase the absorption of Ag cation [27]. Afterward, 200 µL of the prepared Ag_2_CO_3_ solution was dropped onto the CF substrate via manual pipetting. The film was first heated in an oven at 70 °C for 30 min to remove residual methanol. Subsequently, the film was thermally treated at 130 °C for 30 min to form a thin Ag layer. The resulting film consisted of three layers including a glass slide, CF, and Ag layer, with a size of 24 × 40 × 2 mm (Figure 1b). This simple fabrication method can potentially be applied to other CF-based composite materials [28].

The thermal cycler consisted of the Ag/CF film, a cooling fan, and control modules. The cooling fan was mounted on the side of the thermal cycler and was also controlled by a MOSFET (Appendix A). For accurate temperature measurement, a non-contact temperature sensor was attached to the bottom of the film. In addition, the relay module was configured to control the input voltage of the film by pulse-width modulation (PWM). A proportional–integral–differential (PID) controller was used to adjust the PWM and control the temperature of the film. All components were assembled using 3D-printed materials, and the total size of the thermal cycler was 71 × 115 × 76 mm (Appendix A). A custom-written Python script running on the Raspberry Pi was used to control the thermal cycler. The total cost for assembly of our thermal cycler was less than USD 200, and the total power consumption was about 5 W. These results demonstrated the significant advantages of our thermal cycler, compared with commercially available systems (Appendix A).

### 3.2. Characterization of Ag/CF Film

To demonstrate the electrochemical deposition of Ag on a CF surface, SEM and EDS analyses were conducted. In SEM images, the homogeneous Ag layer was observed in the Ag-coated CF, while the untreated CF showed a clean straight layer, with a diameter of 6.9 μm (Figure 2a,b). The elemental analysis of EDS revealed that the atomic percentage of the silver was increased (~23.3%) in the Ag-coated CF, while the untreated CF contained 100% of carbon. These results confirmed the successful electrodeposition of Ag on a CF surface (Figure 2c,d). To determine the optimal concentration of Ag_2_CO_3_ solution, morphological and electrothermal analyses were conducted. Figure 3a presents SEM images of the Ag-coated CF with concentrations of 150, 300, 450, 600, and 750 mM. With increasing concentrations of the Ag_2_CO_3_ solution, the silver particles continued to grow until they eventually covered the CF surface. EDS mapping images showed that the area covered by Ag was increased with Ag_2_CO_3_ concentrations (Figure 3b). As shown in Figure 3c, the area fraction of Ag was measured. The fraction of area was significantly increased from 25.3% at 150 mM to 78% at 300 mM and converged to 99% at 600 mM. The weight gain of Ag was also increased from 15.7% at 150 mM to 59.2% at 300 mM and converged to 88% at 600 mM (Figure 3d). These results confirmed that a CF surface could fully be covered with Ag using 600 mM of Ag_2_CO_3_ solution.

### 3.3. Electrothermal Properties of Ag/CF Film

To investigate the effect of the concentration of Ag_2_CO_3_ solution on the electrical properties of the film, sheet resistance was measured. As shown in Figure 4a, sheet resistance was inversely proportional to concentrations and converged to 0.7 Ω/sq at 600 mM, showing that Ag coverage with high electrical conductivity (6.3 × 10^7^ m/Ω) was increased. The current corresponding to the increasing DC voltage from 0.2 V to 1 V was measured and plotted on I–V curves (Figure 4b). The results revealed that the current of the film was linearly increased with the applied voltage, indicating the stable resistance and electric properties of the film in all concentrations. We also tested the thermal stability of the Ag/CF films (Figure 4c). With increasing temperature, the total resistance of the film remained constant in all concentrations. To further investigate the thermal effects on the Ag/CF film, TGA measurements were performed (Appendix A). A negligible weight loss (0.14%) was observed at the denaturation temperature of PCR (95 °C). More than 96% of the weights remained at 600 °C, showing that the Ag/CF film can be utilized to perform stable PCR thermal cycling.

Furthermore, the heating performance of the Ag/CF film was examined (Figure 5). A constant DC voltage was applied to the Ag/CF film with different Ag_2_CO_3_ concentrations (Figure 5a). Heating rates were calculated by measuring the time to reach denaturation temperature (95 °C) from room temperature. Up to a concentration of 600 mM, the heating rate was increased to 4.8 °C/s as the concentration of Ag_2_CO_3_ increased. Given the electrical properties and heating performance, we selected 600 mM as the optimal concentration of Ag_2_CO_3_ solution for PCR applications. An increasing voltage ranging from 0.2 V to 1.0 V was applied to validate the relationship between the heating rate and the applied voltage (Figure 5b). The heating rate showed a square proportion with the applied voltage, which could be explained by the Joule’s law ($Q=\frac{V^{2}}{R}t$, where Q is the heating value of the film, V is the input voltage, R is the resistance, and t is the operating time) [29]. To further evaluate the temperature homogeneity of the Ag/CF film, the temperature was simultaneously measured at three different points (A, B, C) (Figure 5c). The temperature was gradually increased from 40 °C to 100 °C, and the temperature was measured every 10 °C (Figure 5d). As shown in the thermal image, the low-temperature variance across the heating area was investigated. From the temperature profile, the average temperature difference was found to be 0.6 °C, indicating a uniform temperature distribution over the heating area. These distinctive properties pave the way toward considering Ag/CF films as potential candidates for various biomedical applications.

### 3.4. PCR Applications

To demonstrate PCR applications of our Ag/CF-film-based thermal cycler, the amplification of the human coronavirus DNA was performed (Figure 6). DNA samples were pipetted onto the PDMS chip and covered with mineral oil to prevent water evaporation during thermal cycling (Figure 6a). For multiplexed detection, three different fluorochromes (FAM, TAMRA, and Cy5) were used for three types of DNA—namely, OC43, 229E, and NL63 (Figure 6b). Using our thermal cycler, 35 cycles of PCR thermocycling were accomplished within 10 min, and the total run time was shortened up to 10-fold, compared with a commercial benchtop PCR thermal cycler (Figure 6c). During thermal cycling, the average heating rate was 4.5 °C/s, and no significant difference was observed in heating performance (Appendix A). The amplification of DNA fragments was demonstrated via agarose gel electrophoresis (Figure 6d). Here, the target DNA fragments showed different base pair sizes (64, 121, and 174 bp) to validate the specificity of the assay. As shown in the image, the amplification products showed clear bands, while none of the template control (NTC) samples resolved any amplicons. Additionally, each sequence of the product was identical to the base pair size of the corresponding gene, suggesting that the target DNA could be successfully amplified by our thermal cycler. Afterward, PCR products were analyzed using fluorescence signals. The fluorescence images were acquired with a fluorescence microscope, and the images were subsequently merged into a single image. The fluorescence image showed a distinct fluorescence signal corresponding to each fluorescence probe, and the three types of human coronavirus could be distinguished by fluorescence intensities (Figure 6e,f). Therefore, we confirmed that the multiplexed detection of human coronaviruses was successfully demonstrated with excellent specificity.

## 4. Conclusions

In this paper, we presented a conductive Ag/CF-film-based portable thermal cycler for rapid detection of human coronavirus. The Ag/CF film was simply fabricated by electrodeposition of Ag on the CF surface. The successful coating of Ag and the electrical/thermal properties were demonstrated with SEM and EDS analyses. Under optimal conditions, the Ag/CF film showed excellent electrical conductivity (0.7 Ω/sq), resulting in comparable heating performance (~4.5 °C/sec) with low power consumption (<5 W). The thermal cycler consisted of a Ag/CF film, cooling fan, control module, and firmware, with a portable size (71 × 115 × 76 mm). Using our thermal cycler, the multiplexed amplification was successfully demonstrated. The results showed that 35 cycles of PCR were conducted within 10 min, and three types of human coronavirus (OC43, 229E, and NL63) could be simultaneously detected by distinct fluorescence signals. The fluorescence-based detection showed the potential of our thermal cycler to be utilized for quantitative analysis by coupling with a real-time monitoring setup [30]. These practical applications can pave the way for their application in various biomedical applications. In the future, combining complementary metal-oxide-semiconductor (CMOS) sensors used for fluorescence imaging would further enhance POCT capability [31]. In addition, our thermal cycler could be used for digital PCR assay by utilizing digitation methods, such as microwells [32,33], droplets [34], and microchannels [35]. Therefore, Ag/CF-film-based thermal cyclers can potentially be useful in the onsite detection of infectious diseases in resource-limited settings.

## Figures and Tables

**Figure 1 polymers-14-01983-f001:**
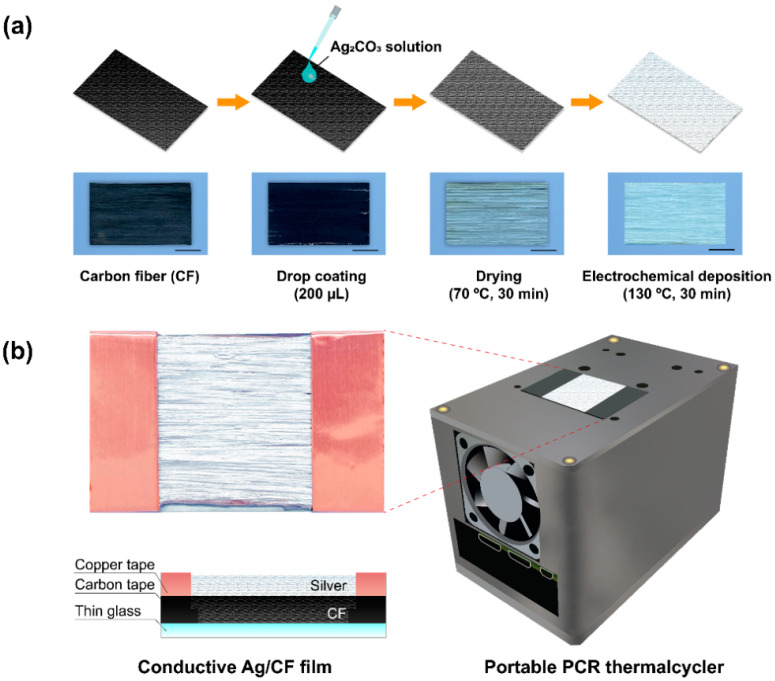
Schematic of Ag/CF film-based portable thermal cycler: (**a**) the fabrication process of Ag/CF film; (**b**) schematic drawing of the thermal cycler and cross-sectional view of the fabricated Ag/CF film. The scale bar is 10 mm.

**Figure 2 polymers-14-01983-f002:**
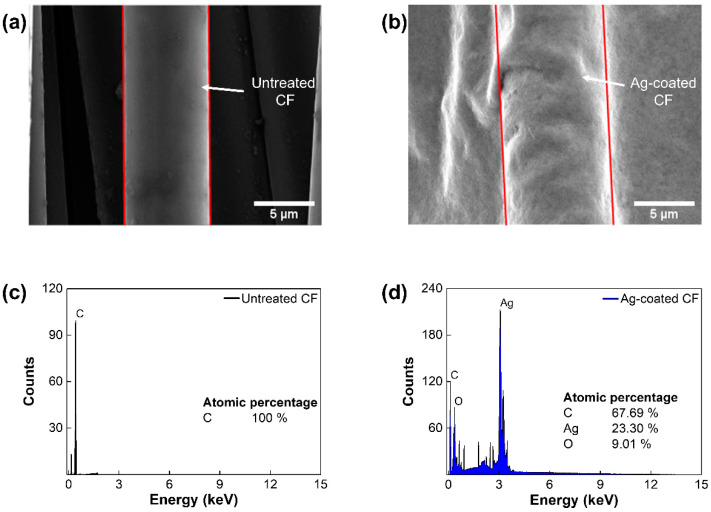
Elemental composition analysis of the Ag/CF film: SEM images of (**a**) untreated CF and (**b**) Ag-coated CF; elemental composition of (**c**) untreated CF and (**d**) Ag-coated CF.

**Figure 3 polymers-14-01983-f003:**
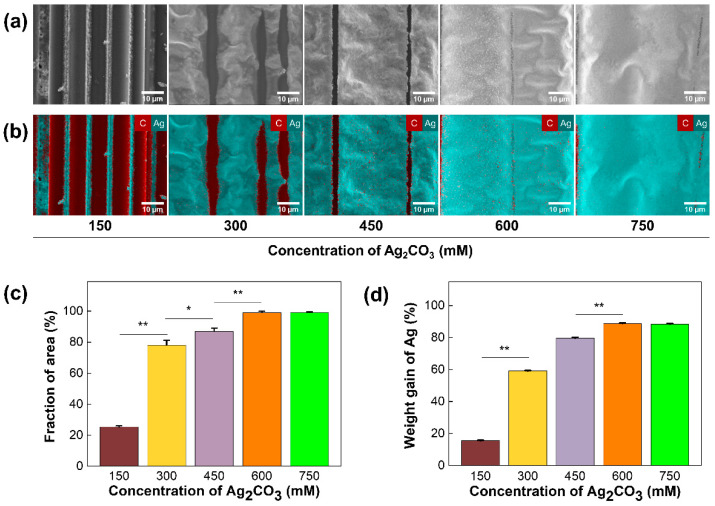
Characterization of the Ag/CF film: (**a**) SEM images and (**b**) EDS mapping images of the Ag/CF film coated with different concentrations of Ag_2_CO_3_ solutions; (**c**) graph representing fraction of area of Ag with respect to Ag_2_CO_3_ concentrations. (**d**) Graph representing weight gain of Ag with respect to Ag_2_CO_3_ concentrations (* *p* < 0.05, ** *p* < 0.01).

**Figure 4 polymers-14-01983-f004:**
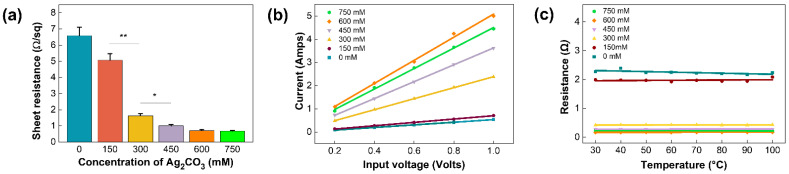
Electrical properties of the Ag/CF film: (**a**) sheet resistance of Ag/CF film with Ag_2_CO_3_ concentrations (* *p* < 0.05, ** *p* < 0.01); (**b**) I–V characteristics of Ag/CF film with respect to Ag_2_CO_3_ concentrations; (**c**) relationship between total resistance and temperature with various Ag_2_CO_3_ concentrations.

**Figure 5 polymers-14-01983-f005:**
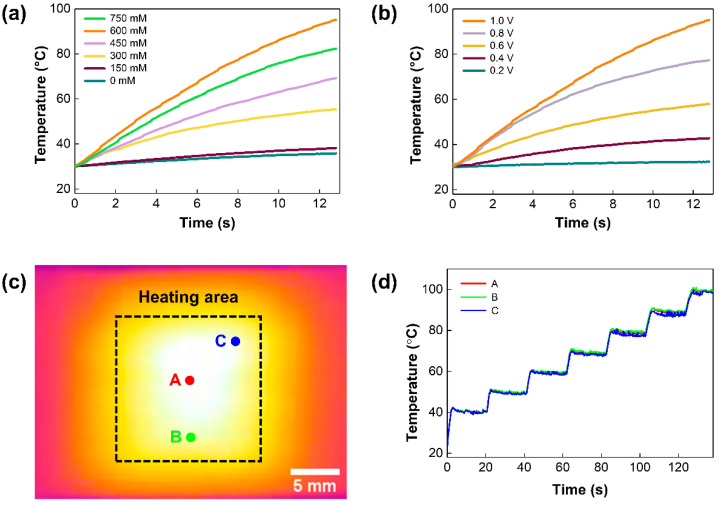
Heating performance of the Ag/CF film: (**a**) electrothermal performance of Ag/CF film with Ag_2_CO_3_ concentrations; (**b**) thermal response of Ag/CF film of 600 mM under various input voltage rates; (**c**) thermal images of heating area at 90 °C; (**d**) temperature profile of three monitoring points. The temperature profile shows homogeneity across the heating area.

**Figure 6 polymers-14-01983-f006:**
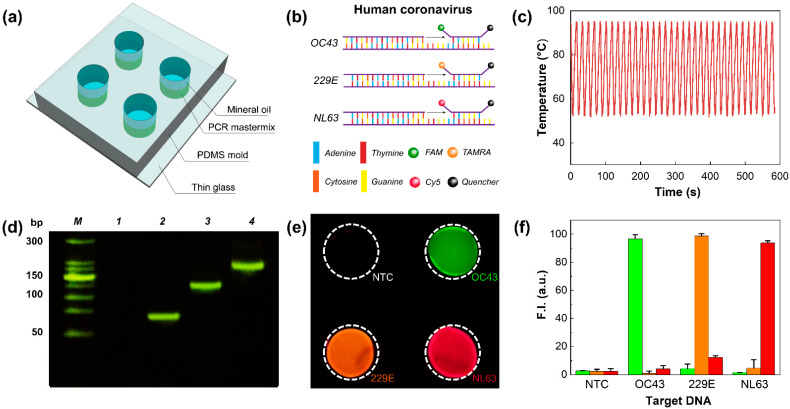
PCR applications of the Ag/CF film-based thermal cycler: (**a**) schematic drawing of multiplexed PCR assay; (**b**) hydrolysis of different TaqMan probes generating distinct fluorescence signals; (**c**) temperature profile of 35 cycles of PCR; (**d**) gel electrophoresis image representing the amplicons. Lane M: 1kb DNA ladder, Lane 1: NTC, Lane 2: OC43, Lane 3: 229E, Lane 4: NL63; (**e**) fluorescence image representing multiplexed PCR assay; (**f**) graph representing fluorescence intensities of the fluorescence intensities.

## Data Availability

The data presented in this study are available on request from the corresponding author.

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
