# Peer review of "Conductive Silver/Carbon Fiber Films for Rapid Detection of Human Coronavirus"

_polymers, 2022, doi:10.3390/polym14101983_

Round 1
Reviewer 1 Report
The paper entitled Conductive silver/carbon fiber film for rapid detection of human coronavirus is very interesting and well written. The author has reported on conductive silver/carbon fiber (Ag/CF)-based PCR thermal cycler with low power consumption (< 5 W), efficient heating (~ 4.5 ºC/sec), low-cost (< $200), and portable size (11.5ï‚´7.1ï‚´7.5 mm). The Ag/CF film, which was used as a heating source of the thermal cycler, was fabricated by the electrochemical deposition method. The successful coating of Ag was characterized by a scanning electron microscope (SEM) and confirmed by en-ergy-dispersive X-ray spectroscopy (EDS). The Ag/CF film showed the excellent electrical/thermal conductivity and durability. Using our thermal cycler, 35 cycles of PCR were accomplished within 10 minutes. We also successfully demonstrated the multiplexed detection of various human coro-naviruses (OC43, 229E, and NL63) using our thermal cycler.
The paper is very well written and presented.
A couple of comments are given below
1: The author state the Conductive silver/carbon is cheaper and portable. It is important to make a comparison of the cost and size of the existing thermal cyclers in the market with the one reported in this paper.
2: it will also be better to provide real images of the process along with the schematic in Figure1.
Reviewer 2 Report
This manuscript describes a conductive silver/carbon fiber (Ag/CF) film-based thermal cycler with an efficient heating rate. The optimized Ag/CF film showed excellent electrical/thermal 70 conductivity, and PCR thermal cycling time was reduced to 10 minutes. However, I did not find anything related to polymer, this manuscript would fit better in journals that focused on sensing, detection or diagnosis.
Reviewer 3 Report
The manuscript is focused on the design of conducting carbon fiber film coated with silver. This film then was used to construct a POCT PCR thermal cycler.
- This work doesn't include any polymer research in it.
-
Reviewer 4 Report
This work is devoted to conductive silver/carbon fiber film for rapid detection of human coronavirus. The Ag/CF film was simply fabricated by electrodeposition of Ag on CF surface, which was characterized by SEM and EDS analysis. The work is of interest because by the developed thermal cycler 3 types of human coronavirus (OC43, 229E, and NL63) could be simultaneously detected within 10 minutes and it could be potentially useful in an onsite detection of the infectious diseases in a resource-limited setting. The article looks like a short communication and may be published after minor revision.
Notes:
- Authors should avoid any abbreviations in the Abstract of the article.
- The letterings should be increased in the Scheme 1. They are difficult for reading.
- Please check the axis name in the Figure 4b. Perhaps it should be named as “current (I)”?
- On the Figure 6d what it means the signals at the left side before NTC? I think they should be denoted.
- For characterization of prepared material especially thermostability properties it would be better to study TG/DSC (Thermogravimetry and Differential Scanning Calorimetry) analysis of Ag/CF film.
Round 2
Reviewer 3 Report
it can be accepted now.
